# Correlation between a Force-Sensing Oral Appliance and Electromyography in the Detection of Tooth Contact Bruxism Events

**DOI:** 10.3390/jcm11195532

**Published:** 2022-09-21

**Authors:** Pietro Maoddi, Edoardo Bianco, Marco Letizia, Matteo Pollis, Daniele Manfredini, Marcello Maddalone

**Affiliations:** 1Aesyra SA, 1015 Lausanne, Switzerland; 2Department of Medicine and Surgery, University of Milano Bicocca, 20900 Monza, Italy; 3Department of Medical Biotechnology, School of Dentistry, University of Siena, 53100 Siena, Italy

**Keywords:** bruxism, electromyography, force sensing, oral appliance, sleep bruxism

## Abstract

Background: Oral appliances embedding sensors can be interesting tools for monitoring tooth contact bruxism in a home environment, as they address some of the usability limitations of portable electromyography (EMG) systems. In this study, an oral appliance for sleep bruxism monitoring was compared to an electromyograph. Methods: Simulated bruxism events with tooth contact, specifically clenching and grinding, and other occlusal activities unrelated to bruxism, were measured in 23 subjects with the two instruments simultaneously. The recordings were analyzed automatically by a computer program in order to compare the two techniques. Results: The two instruments were found to be strongly correlated in terms of detecting events (*r* = 0.89), and estimating their duration (*r* = 0.88) and their intensity (*r* = 0.83). Conclusions: The two techniques were in agreement in measuring event frequency, duration and intensity in the studied group, suggesting that force-sensing oral appliances have the potential to be easy-to-use tools for home monitoring of bruxism, alone or as complements to portable EMGs.

## 1. Introduction

Sleep bruxism is a common condition defined as a masticatory muscle activity during sleep that is characterized as rhythmic (phasic) or non-rhythmic (tonic) [1]. Patients with sleep bruxism can experience damage to teeth and restorations or implants, develop articular problems and suffer from orofacial pain [2,3]. The relationship between these signs and the presence of sleep bruxism is unclear. Metrics to quantify the motor aspects of sleep bruxism—such as the frequency of occurrence and average duration of bruxism events—are distributed as a continuum and should not be crudely used as cut-off points to assess bruxism as a risk factor [4]. The collection of sleep bruxism data in large populations is needed in order to investigate possible correlations with clinical signs that might be present in the affected patients, enabling more holistic and objective assessment methodologies [5,6].

In daily practice, however, the presence and severity of bruxism in patients is generally assessed from self-reports (feeling of facial contraction, sleeping partner hearing grinding sounds) and clinical signs (tooth wear) [7], rather than instrumental measurements. While this kind of qualitative examination is certainly necessary, it is not sufficient for a complete assessment of the condition. The prevalent use in the clinical setting of an exclusively non-instrumental approach can be attributed to the greater cost and deployment complexity of the more advanced techniques that are available today [8,9,10].

A multiple-night polysomnographic (PSG) recording in a sleep clinic—a complex examination which generally includes an electroencephalogram, an electromyogram of the masticatory muscles, an electrocardiogram, respiratory air flow monitoring and audio/video recordings—has been historically considered the standard of reference for the diagnosis of sleep bruxism, with the number of rhythmic masticatory muscles activation (RMMA) events per hour of sleep being the main metric [11,12]. Limitations of this metric, pertaining to its unclear association with symptoms, such as myofascial pain and shortfalls in characterizing the non-rhythmic muscle activation events that can be present in subjects with bruxism (clenching and bracing), have been recognized, suggesting that PSG may provide only a partial picture of the condition [13].

A wide application (i.e., compatible with the high prevalence of sleep bruxism) of PSG is moreover not practical because of the examination cost and duration. To address this problem, several portable electromyographic (EMG) recording devices have been commercialized which allow monitoring sleep bruxism in a home environment, relying on the activity of one of the masseters [14,15], of one of the temporal muscles [16,17] or of both masseters and the heart rate [18,19].

Portable EMG devices are less complex and thus less expensive with respect to PSG and represent an important advancement in enabling the monitoring of sleep bruxism at home [20]. They do, however, present some limitations in terms of comfort and usability, related to (i) the difficulty for a casual user of correctly positioning the adhesive electrodes on the masseter/temporal muscles; (ii) the reliability of the electrodes’ adhesion to the skin for the entire duration of sleep; (iii) the presence of wires or electronic enclosures attached to the face; and (iv) the need to use these devices in combination with plastic bruxism splints to ensure the protection of the teeth, when long-term use is needed.

The use of modified oral appliances, resembling traditional resin splints but embedding sensors that monitor the occlusal pressure, has been proposed for monitoring bruxism [21,22,23,24,25]. Such devices can potentially address some of the usability limitations listed before, as they (i) are easy for the user to place correctly (fit on the dental arch in a precise position); (ii) do not involve in principle wires or enclosures attached to the face (self-contained appliances have been demonstrated); (iii) do not need to be combined with additional devices to protect the teeth. The comfort that is achievable is dependent on the device’s size, which can be reduced by miniaturization of the internal electronics. The correlation between detecting bruxism events through a masseter EMG and a pressure-sensing oral appliance has been previously investigated by researchers [26]. This study differs in that both the temporalis and masseter EMG were used; the oral appliance was a commercial device rather than a custom-made prototype, with capacitive rather than piezoelectric pressure sensors; both bruxism-related and unrelated events were included; and large numbers of subjects and events were studied.

The purpose of this study was to compare a force-sensing oral appliance with an EMG device in detecting simulated bruxism events involving tooth contact. The null hypothesis was that the two instruments are not correlated in terms of detecting (i) the presence, (ii) the duration and (iii) the intensity (exerted force) of the events.

## 2. Materials and Methods

Twenty-three subjects, 13 females and 10 males, average age 39.0 ± 15.2 years, participated in the study. Inclusion criteria were diagnosis of bruxism according to the American Academy of Sleep Medicine criteria [27], age between 18 and 70 years old, willingness to show up at all appointments scheduled by the investigator. Exclusion criteria were TMJ pain, masseter and temporalis pain with functional limitation, more than two missing teeth per hemi-arch and more than one missing molar, patient ongoing orthodontic therapy, patients having maxillary hypoplasia, patient having had recent TMJ trauma or oral or maxillofacial surgical intervention. All subjects signed an informed consent form prior to inclusion in the study. The study was conducted in accordance with the Declaration of Helsinki and approved by the Ethics Committee of ASST Monza (registration code: RCS-MA-001).

The force-sensing oral appliance, AesyBite Discover (ABD, manufactured by Aesyra SA, Lausanne, Switzerland), intended for monitoring the sleep bruxism activity characterized by tooth contact by sensing the occlusal pressure exerted during bruxism events, was used in this study. The ABD oral appliance is a nightguard that is customized by low-temperature thermoforming directly on the upper dental arch of the user (Figure 1a), which makes it deliverable in a single visit. It embeds four pressure sensors in its occlusal surface distributed along the dental arch (Figure 1b) and an electronic module that records the time of occurrence, the duration and the intensity of bruxism events. Such data can be downloaded via near field communication (NFC) to a dedicated mobile application and displayed.

### 2.1. Experimental Protocol

Each participant completed the study in a single session. All the appliance and EMG electrode fitting operations were performed by the same operator.

In the first part of the session, the ABD appliance was customized to the dental arch of the subject with the procedure specified by the manufacturer (softening by immersion in 75 °C water for 2 min, positioning on the maxillary arch using the included thermoforming tray, then instructing the subject to bite with moderate strength for 1 min).

Four electrodes of the TMJOINT system (BTS Bioengineering Corp., Quincy, MA, USA) were placed in related to the left and right temporalis and masseter muscles of the subject. The same EMG device was used for all the subjects; only the disposable adhesive electrodes were changed.

The subject, wearing the ABD oral appliance and the EMG electrodes at the same time, was made to sit upright in front of a computer monitor. Custom software showed a sequence of instructions to be performed for a certain time, and a countdown timer to help the subject in the execution. The actions to be executed in response to the instructions were explained to the subject prior to letting them execute the entire sequence.

The actions of the sequence included 20 bruxism-related events (10 teeth clenching and 10 teeth grinding events) and 20 bruxism-unrelated events that can occur during sleep (5 swallowing, 5 lightly nodding the head, 5 lightly shaking the head, 5 speaking events). Every action was separated by 10 s of inactivity.

Signals from the EMG system were recorded during the sequence execution with the FreeEMG software (BTS Bioengineering Corp., Quincy, MA, USA) using a sampling frequency of 1 kHz. Data recorded by the ABD appliance were downloaded after the sequence execution by reading the appliance’s memory with an NFC-enabled mobile phone (Galaxy A8, Samsung, Seoul, South Korea).

### 2.2. Data Reduction and Analysis

The DC offset from the EMG signals was eliminated by subtracting the mean from each signal. The signals were bandpass filtered (Butterworth, 4th order) between 10 and 400 Hz and rectified by taking the absolute value. A final smoothing pass was performed by applying a moving average with a 2.0 s wide window. Four signals were thus obtained: $x_{RT}$, $x_{LT}$, $x_{RM}$, $x_{LM}$, corresponding, respectively, to the right and left temporalis and to the right and left masseter.

In order to identify categorical events (i.e., detected or not detected events), a binarized signal $x_{B}$ was constructed for each subject by means of a threshold applied to the average between the channels:(1)$$x_{B}=\left\{\begin{array}{c} 1\;\left(Detected\right),\;\;if\bar{x}>th\\ 0\;\left(Undetected\right),\;\;otherwise \end{array}\right.$$
where the signal
(2)$$\bar{x}=\frac{1}{4}\left(x_{RT}+x_{LT}+x_{RM}+x_{LM}\right)$$
is the average of the four EMG channels. This procedure mimics the detection mechanism of the ABD oral appliance, which detects a bruxism episode when a capacitance threshold is exceeded. As inter-subject variability in EMG levels prevents using a predetermined threshold common to all the subjects, the threshold th was manually chosen for each subject in order to correspond to a detection level approximately equivalent to that of the oral appliance. In practice, the threshold was adjusted until a good visual correspondence between the binarized signals of the EMG and of the appliance could be observed. Similarly, the signal
(3)$$\bar{s}=\frac{1}{4}\left(s_{1}+s_{2}+s_{3}+s_{4}\right),$$
computed as the mean of the 4 sensors of the appliance, was considered for ABD. The level of this signal represents the average pressure on the appliance, as ABD records only the time-averaged value for each sensor during an event (and not the whole dynamics of the pressure during the event).

For each instruction given to the subject, an event detection window—starting 5 s before and ending 5 s after the instruction—was considered (Figure 2). In order to reduce bias, the analysis of the events within the detection window corresponding to each instruction (i.e., computation of the event presence or absence and measurement of its duration and intensity) was performed automatically by a custom-made Python script. An example of the automatic analysis process on one subject is represented in Figure 3.

### 2.3. Calculation of the Correlation Coefficients

The signal from the ABD appliance and the EMG binarized signal were checked for events in each detection window, compiling a 2 × 2 contingency table for each subject (Table 1).

For each subject, the Pearson correlation coefficient
(4)$$r_{D,i}=\phi_{D,i}=\frac{\mathrm{TD}\cdot \mathrm{TU}-\mathrm{FD}\cdot \mathrm{FU}}{\sqrt{\left(\mathrm{TU}+\mathrm{FU}\right)\left(\mathrm{TD}+\mathrm{FD}\right)\left(\mathrm{TU}+\mathrm{FD}\right)\left(\mathrm{TD}+\mathrm{FU}\right)}}$$
between ABD and EMG was calculated. In cases where this calculation yielded a perfect correlation (*r* = 1), the value *r* = 0.99 was used instead, to allow computing the Fisher z-transformation. The combined correlation for event detection $r_{D}$ between all subjects was calculated with the Fisher z-transformation method [28,29]—i.e., transforming every coefficient, computing the average, then transforming back.

For the sake of the comparison with the EMG, the following figures of merit referring to event detection were calculated for the ABD appliance: sensitivity = TD/(TD + FU); specificity = TU/(TU + FD); accuracy = (TD + TU)/(TD + TU + FD + FU).

Only events correctly detected by both the EMG and ABD were considered for the event duration and intensity analysis. The uptimes $T_{EMG}$ and $T_{ABD}$ of the EMG and ABD binarized signals (inside the detection window) were considered as the estimated duration for each event, and the Pearson correlation coefficient between them was calculated for each subject. The combined correlation coefficient for event duration $r_{T}$ was calculated with the Fisher z-transformation method, weighted on the number of events considered per subject.

The event intensities $I_{EMG}$, $I_{ABD}$ were obtained by computing the time average of the EMG signal and the ABD signal, respectively, during each event. The combined correlation coefficient for event intensity $r_{I}$ was calculated with the same approach used for event duration.

## 3. Results

The combined correlation coefficient found for event detection, $r_{D}=0.89$, indicates that the event detection by ABD and EMG were strongly correlated. The 920 events considered in total (40 events per subject) were divided, assuming the EMG measurement as the ground truth, into 371 true undetected (TU) events, 450 true detected (TD), 27 false undetected (FU) and 72 false detected (FD) events. The figures of merit for event detection of the ABD appliance found were: sensitivity = 94.3%, specificity = 83.7% and accuracy = 89.2%, indicating generally good agreement with the EMG.

The event duration estimates by the two instruments $T_{EMG}$, $T_{ABD}$ were found to be strongly correlated ($r_{T}=0.88$). The box of plot of their difference $\Delta T=T_{ABD}-T_{EMG}$ is represented in Figure 4. The distribution of ΔT is non-normal (kurtosis ≈ 7) and presents a median of 0.03 s and an interquartile range of 1.34 s. The event intensity estimates, $I_{EMG}$ and $I_{ABD}$, were also found to be strongly correlated ($r_{I}=0.83$).

A summary of the correlations calculated is provided in Table 2. In the majority of subjects, there was a strong correlation between EMG and ABD in detecting events and estimating their duration and intensity (Figure 5), exceptions being subjects 12, 13 and 19 (Table 2), who presented very poor or no correlation in detected events (*r* < 0.4), due to the presence of several falsely detected events.

## 4. Discussion

The purpose of this study was to compare a force-sensing oral appliance with an EMG device in detecting simulated bruxism events involving tooth contact. The two instruments were found to be strongly correlated in terms of detecting the presence of the events ($r_{D}=0.89$), their duration ($r_{T}=0.88$) and their intensity ($r_{I}=0.83$) in the sample studied.

It was observed that the average difference in the event duration estimate between EMG and ABD was negligible (0.22 s). An exact correspondence was not expected because the duration of a detected EMG event is known to depend on the detection threshold considered [30], and the ABD appliance has a predefined threshold. Part of the dispersion in the data can be attributed to the fact that the ABD appliance records event durations with a 1 s resolution. These results suggest that while the ABD oral appliance does often underestimate or overestimate the duration of single events with respect to the EMG, the errors cancel out for a sufficient number of events, and there is good agreement between the two techniques in estimating the average event duration. While correlations in event duration for specific event types (e.g., grinding and clenching) were not calculated in this study, the duration correlation found over all events, $r_{T}=0.88$, falls in the range of those reported by Takeuchi et al. [26], which were *r* = 0.79 and *r* > 0.99 for grinding and clenching events, respectively, when comparing an EMG with a sensorized oral appliance.

The correlation between the event intensities measured by the two instruments is a surprising finding, given the relative simplicity of the intensity measures considered (time average of the EMG signal over both the masseters and temporalis, and time averaged capacitance of all ABD sensors) and the non-linear response expected from capacitive sensors.

The fact that the falsely detected events (7.8% of the total) are more numerous than the falsely undetected ones (2.9% of the total) is likely due to the pressure sensors of the ABD oral appliance occasionally remaining in a compressed state for some time after an intense event, resulting in a false detection in the immediately following window. Due to the presence of several falsely detected events, three subjects presented poor or no correlation. The reason for this is unclear. It might have been due to the capacitance threshold of the appliance being too low for the forces exerted by these subjects, possibly leading the electronics to detect the sensors as still pressed after a previous intense event has occurred.

Similar issues with detecting intense clenching events were observed in other sensorized oral appliances [26,31], although based on a different sensing principle (piezoelectric). Further analysis is needed to understand the exact circumstances of occurrence of these false positives and test whether using a less sensitive threshold setting can improve the outcome.

A limitation of this study is that it involved simulated events performed by awake subjects, rather than genuine events recorded in sleeping subjects. Additional studies, involving recordings of sleeping bruxers and non-bruxers and comparisons with validated diagnostic instruments, are necessary to investigate whether a diagnostic application of the ABD appliance can be justified. Moreover, the circumstances under which the appliance appears to be overly sensitive in detecting events should be investigated in order to improve the understanding of the lack of correlation in certain subjects.

The correlations observed in this study might not be evident in a multi-day monitoring period where the EMG levels variability is expected to increase (because of changes in skin hydration, facial hair growth, electrode positioning and adhesion), thereby affecting the event duration and intensity measurements. The ABD device is expected to suffer less from this variability, as its detection principle is based on the occlusal pressure exceeding a fixed threshold, though further testing is necessary for this assessment.

Force-sensing oral appliances, such as the one studied in this work, could be useful tools in the instrumental assessment of sleep bruxism in the home environment. They could be easier-to-use or more comfortable options for patients not compliant with portable EMG monitors, or constitute one of the data sources in multidimensional assessments such as the STAB [5,6].

## 5. Conclusions

The null hypothesis that the EMG and ABD instruments are not correlated in terms of detecting the presence, the duration and the intensity (exerted force) of simulated bruxism events was rejected. The correlation between EMG and ABD observed in the studied group suggests that the two techniques are generally concordant in assessing the characteristics of teeth clenching and grinding events. This correlation suggests that oral appliances such as ABD have the potential to be easy-to-use tools for home monitoring of bruxism, alone or complementary to portable EMGs.

## Figures and Tables

**Figure 1 jcm-11-05532-f001:**
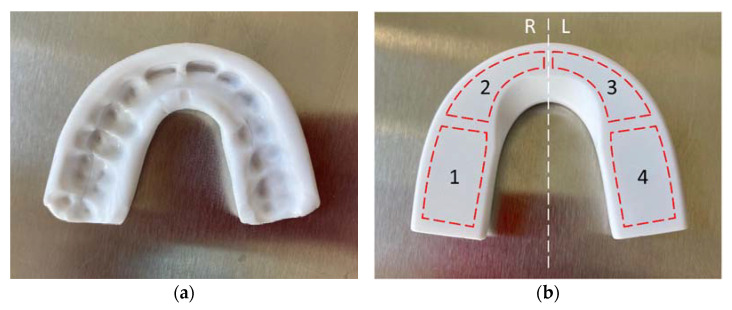
The AesyBite Discover oral appliance. (**a**) Top view showing the maxillary arch impression. (**b**) Occlusal plane view (R: right side, L: left side) with superimposed position of the four pressure sensors (1–4) hidden in the appliance.

**Figure 2 jcm-11-05532-f002:**
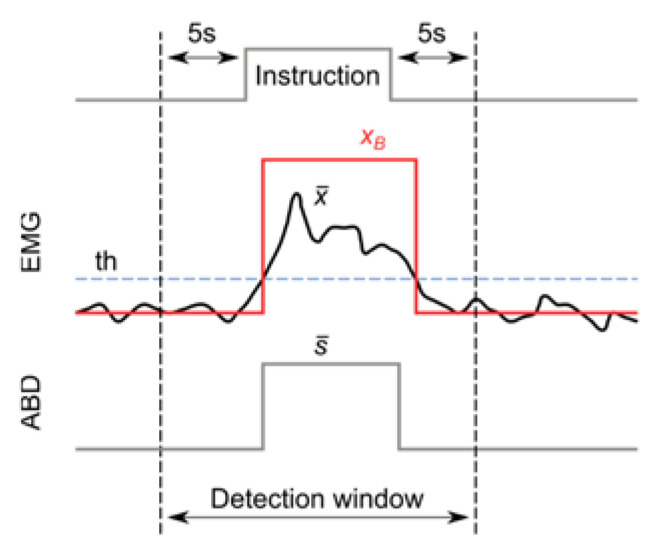
Depiction of the binarized signal $x_{B}$ obtained by thresholding the average EMG signal $\bar{x}$ and of the detection window used for comparing ABD and EMG events.

**Figure 3 jcm-11-05532-f003:**
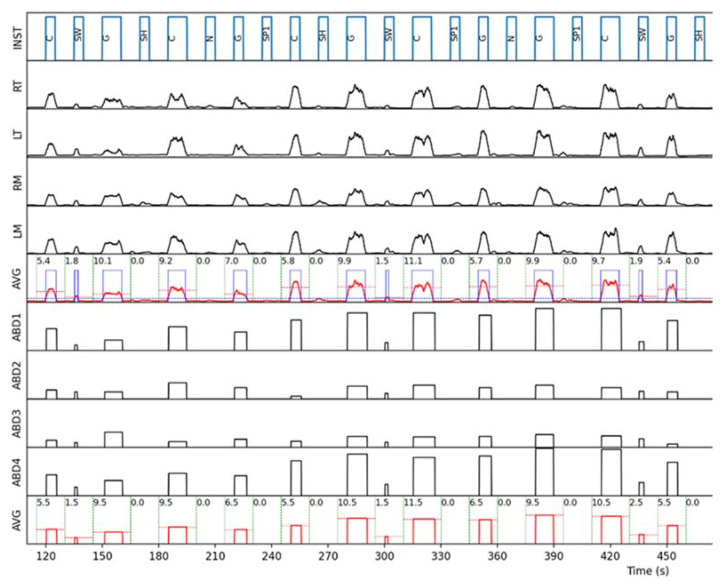
Partial recording of a single subject. The “INST” plot shows the instructions given to the subject: C = clench, G = grind, SW = swallow, SH = shake head, N = nod, SP1 = speak. The “RT”, “LT”, “RM” and “LM” plots show the EMG recordings (RT, LT = right and left temporalis; RM, LM = right and left masseter), and the “AVG” plot immediately underneath shows their average (red) and the automatically detected events (blue binarized signal). The “ABD1–4” plots show the mean pressure detected by each of the four sensors of the oral appliance, and the “AVG” plot immediately underneath shows their average. The numbers inside the detection window edges (delimited by the dotted green lines) represent the computed event duration, and the horizontal dashed lines show the event intensity (time average).

**Figure 4 jcm-11-05532-f004:**
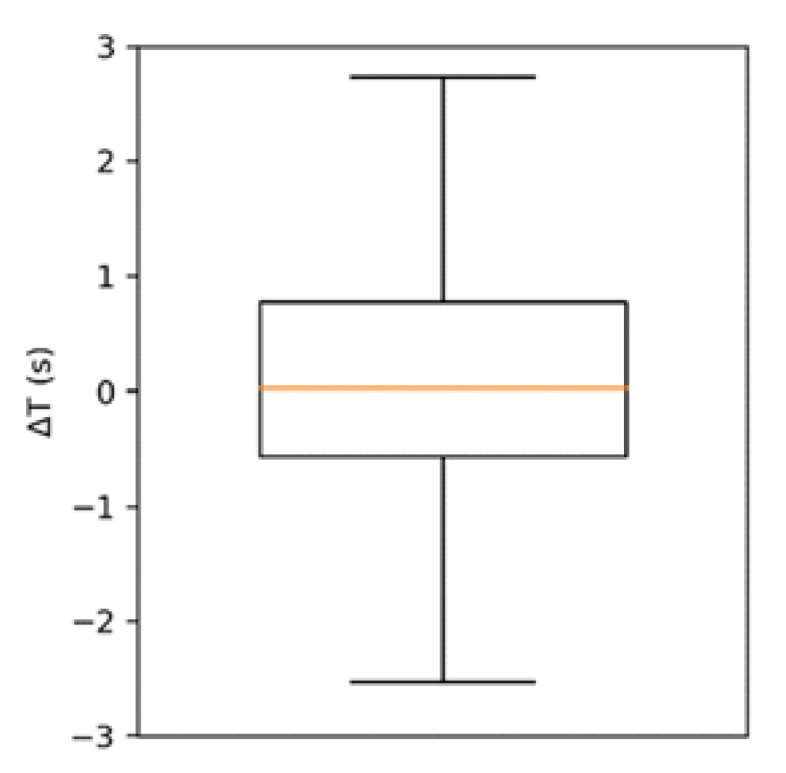
Box plot (outliers not shown) of $\Delta T=T_{ABD}-T_{EMG}$. Average = 0.22 s, median = 0.03 s, interquartile range = 1.34 s.

**Figure 5 jcm-11-05532-f005:**
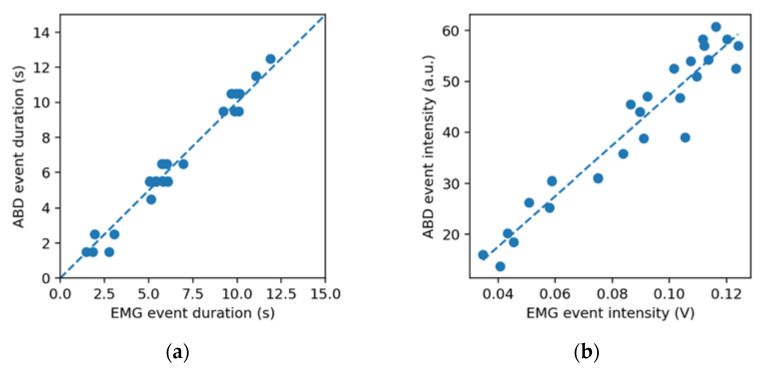
Data from a selected single subject, exhibiting good correlation between the two techniques. (**a**) Scatter plot of the event durations detected by EMG and ABD. The dashed line represents ideal agreement (y = x). (**b**) Scatter plot of event intensity detected by EMG (time averaged voltage) and ABD (value proportional to the time averaged sensor capacitance). The dashed line is the linear regression fit.

**Table 1 jcm-11-05532-t001:** Contingency table used for the analysis of event detection correlation between ABD and EMG, assuming the latter as the ground truth. U = undetected, D = detected, TU = true undetected FU = false undetected, FD = false detected, TD = true detected.

		EMG	
		U	D
ABD	U	TU	FU
	D	FD	TD

**Table 2 jcm-11-05532-t002:** Summary of the results obtained for each test subject.

Subject	$\mathrm{Detection}\;r_{D}$	$\mathrm{Duration}\;r_{T}$	$\mathrm{Intensity}\;r_{I}$
1	0.95	0.59	0.73
2	0.99	0.89	0.98
3	0.82	0.90	0.65
4	0.81	0.79	0.78
5	0.90	0.75	0.59
6	0.85	0.86	0.86
7	0.80	0.92	0.91
8	0.79	0.91	0.82
9	0.99	0.95	0.67
10	0.52	0.45	0.54
11	0.95	0.79	0.90
12	0.35	0.52	0.95
13	0.31	0.50	−0.83
14	0.77	0.77	0.84
15	0.80	0.82	0.62
16	0.95	0.91	0.89
17	0.95	0.41	0.65
18	0.95	0.95	0.54
19	0.00	0.64	0.71
20	0.86	0.97	0.96
21	0.99	0.92	0.74
22	0.99	0.99	0.96
23	0.85	0.96	0.95
Average *	0.89	0.88	0.83

* Computed with the Fisher z-transformation method, weighted on the number of events per subject.

## Data Availability

The data presented in this study are available on request from the corresponding author.

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
