# Peer review of "Correlation between a Force-Sensing Oral Appliance and Electromyography in the Detection of Tooth Contact Bruxism Events"

_jcm, 2022, doi:10.3390/jcm11195532_

Round 1

Reviewer 1 Report

pg 85: how were the 23 patients selected? The statistics are ok with the 23 subjects? Did you calculate the number needed to treat? NNT? If so can we extrapolate the data according to those 23 subjects?

pg 94: please provide the number of the ethics committee

148-151: please provide more explanations regarding the manually chosen threshold

255-265: this kind of events can happen with different digital devices/sensors. Hence the need to compare the results of this particular device with other devices which already exist - in the discussion section

in all the discussion section you should include more references in order to find other articles which deal with similar or different results 

please rephrase the conclusions, the first phrase in particular.

Author Response

Dear reviewer,

Thank you for the time and effort you have dedicated to review the manuscript and for the insightful comments and suggestions you provided.

Please find below our point-by-point responses.

Comment 1: pg 85: how were the 23 patients selected? The statistics are ok with the 23 subjects? Did you calculate the number needed to treat? NNT? If so can we extrapolate the data according to those 23 subjects?

Response: We appreciate the concerns underlying the questions and we are happy to give additional information on the recruitment.

All participants were volunteers compliant with the study inclusion and exclusion criteria described in lines 85-94, and included patients under care at the ASST Monza San Gerardo Odontostomatology department as well as dental medicine students under training in the same department. All participants underwent dental examination and received recommendations for follow-up as appropriate; all were aware of their bruxism condition and most were habitually using a nightguard. No NNT calculation was necessary because no intervention was administered as part of the study.

Regarding the statistics: the null hypothesis of no correlation (H0: r = 0) can be tested with the t-statistic from information given in the article. The average correlations over the 23 subjects for detecting events (r = 0.89), estimating their duration (r = 0.88) and their intensity (r = 0.83) are all significant with p < .05:
r = 0.89, t = 8.9448, p < .001
r = 0.88, t = 8.4903, p < .001
r = 0.83, t = 6.8193, p < .001

Comment 2: pg 94: please provide the number of the ethics committee

Response: We had already provided the Ethics Committee approval number in the “Institutional Review Board Statement” (line 307) in accordance with the article template of the Journal of Clinical Medicine. We now added it to line 94 as well.

Comment 3: 148-151: please provide more explanations regarding the manually chosen threshold

Response: We agree that the explanation can be improved.

The oral appliance detects a bruxism episode when any of its force sensors exceeds a fixed capacitance value, i.e. it uses a predetermined threshold value independent from the user. It is not possible to meaningfully replicate this behavior with the EMG by setting a fixed threshold value (say 1 mV) because inter-subject EMG levels can be very different.

Hence the value of the threshold th was adjusted manually until the shape of the signal  visually resembled the signal , under the assumption that this constitutes a good approximation of setting both devices to an equivalent threshold level.

We have amended the text to explain this procedure more clearly.

Comment 4: 255-265: this kind of events can happen with different digital devices/sensors. Hence the need to compare the results of this particular device with other devices which already exist - in the discussion section

Comment 5: in all the discussion section you should include more references in order to find other articles which deal with similar or different results 

Response: Agreed to both suggestions. We have added some discussion and references to the most similar studies.

Comment 6: please rephrase the conclusions, the first phrase in particular.

Response: We have reviewed the conclusions but it is not clear to us what the specific concern is. The first phrase “The null hypothesis that the EMG and ABD instruments are not correlated [...] is rejected” directly refers to the study purpose and null hypothesis described in the article introduction, lines 80-83.

We hope we have addressed correctly your comments and are available to respond to any further questions and comments you may have.

Best regards,

Dr. Pietro Maoddi and Dr. Edoardo Bianco

Reviewer 2 Report

Dear Authors,

congratulations on yet another interesting article.

I only have a few minor comments:

1) please correct spelling/grammar mistakes in the following lines: 39-40; 73; 75 (investigated by); 100 (makes); 280-281;

2) line 65: in what cases the long term use is  needed?

3) a great number of your citations is auto citations, I understand in most cases, however in the matters like citation no. 8 and similar please provide some other sources as well 

4) by how much the used oral appliance usually increases patients’ VOD?

Author Response

Dear reviewer,

Thank you for the time and effort you have dedicated to review the manuscript and for the insightful comments and suggestions you provided.

Please find below our point-by-point responses.

Comment 1: please correct spelling/grammar mistakes in the following lines: 39-40; 73; 75 (investigated by); 100 (makes); 280-281;

Response: Thank you for pointing out the mistakes, we have corrected them.

Comment 2: line 65: in what cases the long term use is needed?

Response: For example, in cases in which EMG monitoring is used as part of a biofeedback treatment system, such as in the Grindcare device. This EMG-based device is used for 4-6 weeks during which it can be combined with other bruxism management methods including dental splints. [see: Lobbezoo et al. "Consensus‐based clinical guidelines for ambulatory electromyography and contingent electrical stimulation in sleep bruxism." Journal of Oral Rehabilitation 47.2 (2020): 164-169]

Comment 3: a great number of your citations is auto citations, I understand in most cases, however in the matters like citation no. 8 and similar please provide some other sources as well

Response: Agreed. We have added some other sources as suggested.

Comment 4: by how much the used oral appliance usually increases patients’ VOD?

Response: Interesting question, unfortunately we did not measure this parameter during the study and hence we are unable to provide statistics of the VDO increase on the study subjects. In our experience however, the typical VDO increase while wearing the ABD appliance is around 1 mm.

We hope we have addressed correctly your comments and are available to respond to any further questions and comments you may have.

Best regards,

Dr. Pietro Maoddi and Dr. Edoardo Bianco

Round 2

Reviewer 1 Report

Thank you for the modifications! Good luck with your future studies and the improvement of the weak points!